

# Enhancing breast cancer prediction through stacking ensemble and deep learning integration

Fatih Gurcan

Department of Management Information Systems, Faculty of Economics and Administrative Sciences, Karadeniz Technical University, Trabzon, Turkey

## ABSTRACT

Breast cancer is one of the most common types of cancer in women and is recognized as a serious global public health issue. The increasing incidence of breast cancer emphasizes the importance of early detection, which enhances the effectiveness of treatment processes. In addressing this challenge, the importance of machine learning and deep learning technologies is increasingly recognized. The aim of this study is to evaluate the integration of ensemble models and deep learning models using stacking ensemble techniques on the Breast Cancer Wisconsin (Diagnostic) dataset and to enhance breast cancer diagnosis through this methodology. To achieve this, the efficacy of ensemble methods such as Random Forest, XGBoost, LightGBM, ExtraTrees, HistGradientBoosting, AdaBoost, GradientBoosting, and CatBoost in modeling breast cancer diagnosis was comprehensively evaluated. In addition to ensemble methods, deep learning models including convolutional neural network (CNN), recurrent neural network (RNN), gated recurrent unit (GRU), bidirectional long short-term memory (BILSTM), long short-term memory (LSTM) were analyzed as meta predictors. Among these models, CNN stood out for its high accuracy and rapid training time, making it an ideal choice for real-time diagnostic applications. Finally, the study demonstrated how breast cancer prediction was enhanced by integrating a set of base predictors, such as LightGBM, ExtraTrees, and CatBoost, with a deep learning-based meta-predictor, such as CNN, using stacking ensemble methodology. This stacking integration model offers significant potential for healthcare decision support systems with high accuracy, F1 score, and receiver operating characteristic area under the curve (ROC AUC), along with reduced training times. The results from this research offer important insights for enhancing decision-making strategies in the diagnosis and management of breast cancer.

# INTRODUCTION

Breast cancer is one of the most common cancer types affecting women, recognized globally as a significant public health issue that impacts millions of women annually. According to the World Health Organization (WHO), approximately 2.3 million women worldwide are diagnosed with breast cancer each year (*Zhu et al., 2023*). This high incidence

Corresponding author
Fatih Gurcan, fgurcan@ktu.edu.tr

rate makes breast cancer the most prevalent cancer type among women and imposes a substantial burden on global healthcare systems. The increasing number of breast cancer cases underscores the critical importance of early detection, significantly enhancing the effectiveness of treatment processes and patient survival rates (*Fatima et al., 2020*; *Ebrahim, Sedky & Mesbah, 2023*; *Zhu et al., 2023*). Early diagnosis allows for the detection and treatment of cancer at its earliest stages, thereby substantially improving patients' quality of life and chances of survival. However, traditional diagnostic methods are often invasive and time-consuming, requiring procedures such as biopsies and advanced imaging techniques (*Montazeri et al., 2016*). These processes can be costly and burdensome for both patients and healthcare systems. Moreover, the accuracy and timely delivery of results of these methods are critically important for rapidly progressing diseases like breast cancer (*Abreu et al., 2016*; *Montazeri et al., 2016*; *Fatima et al., 2020*).

In this context, machine learning and deep learning technologies offer significant potential in healthcare by providing rapid, accurate, and non-invasive diagnostic capabilities (*Abreu et al., 2016*; *Montazeri et al., 2016*; *Ferroni et al., 2019*; *Fatima et al., 2020*). Machine learning algorithms learn from large datasets, enabling more precise and faster diagnoses in clinical decision support systems. Particularly, deep learning techniques excel in image processing and classification, potentially revolutionizing early diagnosis of diseases like breast cancer (*Lou et al., 2020*; *Al-Azzam & Shatnawi, 2021*; *Naseem et al., 2022*). Ultimately, the integration of ensemble learning and deep learning technologies not only enhances accuracy in breast cancer diagnosis but also accelerates diagnostic processes (*Fatima et al., 2020*; *Karadeniz, Tokdemir & Maraş, 2021*; *Naseem et al., 2022*; *Ebrahim, Sedky & Mesbah, 2023*; *Gurcan et al., 2023*). In recent years, artificial intelligence technologies, particularly in machine learning and deep learning, have been the focus of various significant studies for breast cancer diagnosis (*Montazeri et al., 2016*; *Rasool et al., 2022*; *Ebrahim, Sedky & Mesbah, 2023*). The number of studies conducted to enhance breast cancer diagnosis using various predictive models based on machine learning has recently increased a great deal (*Fatima et al., 2020*; *Abbas et al., 2021*; *Al-Azzam & Shatnawi, 2021*; *Wu & Hicks, 2021*; *Swassen & Okba, 2022*; *Naseem et al., 2022*; *Gurcan, 2023a*). The significant growth in the number of these experimental studies has enabled the development and implementation of more effective strategies in the diagnosis and treatment of breast cancer (*Abreu et al., 2016*; *Ferroni et al., 2019*; *Naseem et al., 2022*).

Efforts have been made to improve the prediction of breast cancer by enhancing the prediction performance of proposed machine learning models; however, there is still potential to create interpretable alternative stacking ensemble and deep learning models to improve prediction performance and overcome limitations. Since there are a number of research focusing on cancer diagnosis using classical machine learning, this study proposes a novel methodology for the use of ensemble learning models (including boosting and bagging algorithms) and deep learning models both alone and integrated with stacking ensemble methods. In light of this context, the objective of this paper is to fill these gaps and improve breast cancer diagnosis through the integration of ensemble and deep learning models utilizing stacking ensemble techniques. The experimental analysis of this study includes ensemble learning, deep learning, and stacking ensemble learning models, as

well as feature selection methods with different backgrounds. Consequently, the research questions (RQ) guiding this study can be summarized as follows:

RQ1. What descriptive features are important in breast cancer diagnosis?

RQ2. What is the efficiency of ensemble learning models in breast cancer diagnosis?

RQ3. What is the efficiency of deep learning models in breast cancer diagnosis?

RQ4. What is the efficiency of stacking ensemble methods that integrate ensemble and deep learning models in breast cancer diagnosis?

The remaining sections of this article are organized as follows. "Related Work" covers previous work. "Materials and Methods" discusses the study's methodology in detail. "Experimental Results and Discussions" offers the study's results from experiments as well as an exhaustive discussion. Key findings and upcoming research are finally summarized in "Conclusions".

## BACKGROUND AND RELATED WORK

This section will review and present the literature on breast cancer detection based on the use of machine learning techniques. Machine learning (ML) encompasses a range of algorithms and models that can learn from data and make predictions or decisions without being explicitly programmed (_Raschka & Mirjalili, 2019_; _Gurcan et al., 2022a_; _Gurcan, 2023a_; _Nelli, 2023_). Supervised and unsupervised learning are two fundamental approaches in machine learning (_Gurcan et al., 2022b_; _Nelli, 2023_). Supervised learning trains models using labeled data, where each input has a corresponding correct output value (_Pedregosa et al., 2011_; _Al-Azzam & Shatnawi, 2021_). This method learns from past experiences to make predictions on new data, with examples including regression and classification algorithms (_Ferroni et al., 2019_; _Rahman et al., 2021_; _Wu & Hicks, 2021_). Unsupervised learning, on the other hand, uses unlabeled data to discover hidden patterns, groups, or structures within the dataset (_Aurélien, 2019_; _Gurcan, 2024_; _Wu, Nguyen & Luu, 2024_). Clustering, topic modeling and dimensionality reduction methods fall into this category (_Gurcan et al., 2022a_; _Gurcan, 2023b_; _Wu, Nguyen & Luu, 2024_). The primary advantage of supervised learning is its ability to make highly accurate predictions for specific tasks, thanks to training based on labeled datasets (_Plotnikova, Dumas & Milani, 2020_; _Al-Azzam & Shatnawi, 2021_; _Nelli, 2023_). In recent years, machine learning and deep learning techniques have been used in numerous studies for the detection and classification of breast cancer, yielding significant insights (_Abreu et al., 2016_; _Fatima et al., 2020_; _Chugh, Kumar & Singh, 2021_). Applications of ML in breast cancer research have led to the development of various predictive models, each with its own strengths and weaknesses (_Abreu et al., 2016_; _Chugh, Kumar & Singh, 2021_; _Naseem et al., 2022_). Early studies predominantly utilized traditional machine learning algorithms such as logistic regression, support vector machines (SVM), k-nearest neighbors (k-NN), Naive Bayes, and decision trees for breast cancer prediction (_Fatima et al., 2020_; _Al-Azzam & Shatnawi, 2021_; _Rasool et al., 2022_; _Ebrahim, Sedky & Mesbah, 2023_). Subsequent research integrated advanced ensemble methods like Random Forest, GradientBoosting, XGBoost, and LightGBM into research methodologies (_Raschka & Mirjalili, 2019_; _Abbas et al., 2021_; _Naseem et al., 2022_; _Ebrahim, Sedky & Mesbah, 2023_).

Additionally, over the past decade, numerous studies have examined the effectiveness of deep learning models such as convolutional neural network (CNN), recurrent neural network (RNN), and long short-term memory (LSTM) in breast cancer diagnosis (*Gulli & Pal, 2017*; *Aurélien, 2019*; *Chugh, Kumar & Singh, 2021*; *Rahman et al., 2021*; *Din et al., 2022*; *Ebrahim, Sedky & Mesbah, 2023*). A comparative analysis study using machine learning models for breast cancer diagnosis was presented by *Ebrahim, Sedky & Mesbah (2023)*. A dataset of 1.7 million data records was received from the US National Cancer Institute (NCI) for this investigation. The accuracy evaluation covered both deep learning and conventional approaches. Several classical methods were used, such as decision tree (DT), SVM, ensemble techniques (ET), linear discriminant (LD), and logistic regression (LR). Techniques including the RNN, deep neural network (DNN), and probabilistic neural network (PNN) were used as a comparison. The impact of feature selection on accuracy was also examined in this study. The findings showed that ensemble techniques and decision trees achieved superior performance compared to other methods, reaching an accuracy of 98.7% (*Ebrahim, Sedky & Mesbah, 2023*).

Tumor type predictions were made in the study paper by *Ak (2020)* using information on breast tumors gathered by Dr. William H. Walberg from the University of Wisconsin Hospital. The dataset was analyzed using a range of machine learning algorithms and data visualization techniques, focusing on improving classification accuracy and identifying key patterns within the data. The results achieved from the logistic regression model, which included all features, revealed the maximum classification accuracy (98.1%), indicating an improvement in accuracy performance with the proposed approach (*Ak, 2020*).

*Rasool et al. (2022)* introduced data exploration techniques (DET) to improve the precision of breast cancer diagnosis and developed four distinct prediction models. Before building the models, they applied a four-layer DET process that included feature distribution analysis, correlation assessment, feature elimination, and hyper parameter tuning on the Breast Cancer Wisconsin (Diagnostic) dataset (BCWD) and Breast Cancer Coimbra Dataset (BCCD). This approach led to improved diagnostic performance, with polynomial SVM reaching 99.3% accuracy, logistic regression 98.06%, k-nearest neighbors 97.35%, and the ensemble classifier 97.61% on the BCWD dataset (*Rasool et al., 2022*).

*Naseem et al. (2022)* conducted a study using artificial neural network (ANN) and various classifier ensembles for automatic breast cancer (BC) diagnosis and prognosis detection. They evaluated different ensemble approaches alongside other variations of machine learning models, both with and without the use of oversampling, across two standard datasets. Their findings showed that the ensemble technique surpassed other advanced methods, reaching an accuracy of 98.83% (*Naseem et al., 2022*).

For radial basis function kernel extreme learning machines (RBF-KELM), *Swassen & Okba (2022)* developed an expert system based on an evolutionary technique called differential evolution (DE) for breast cancer diagnosis. To verify the effectiveness of the proposed approach, DE-RBF-KELM was examined on two datasets: The Mammographic Image Analysis Society (MIAS) and the Breast Cancer Wisconsin (Diagnostic) (BCWD) dataset, achieving satisfactory results compared to traditional approaches (*Swassen & Okba, 2022*).

*Wu & Hicks (2021)* selected characteristics (genes) used in creating and verifying classification models by using RNA sequencing data from 992 non-triple-negative and 110 triple-negative breast cancer tumor samples from The Cancer Genome Atlas. Using characteristics chosen at various threshold levels to train models for categorizing the two forms of breast cancer, they assessed four different supervised models: Naïve Bayes, support vector machines, k-nearest neighbor, and decision tree. The suggested techniques were tested and validated using independent gene expression datasets. Of the ML algorithms that were assessed, the support vector machine approach performed better than the other three in reliably classifying breast cancer as either non-triple-negative or triple-negative, with less misclassification errors (*Wu & Hicks, 2021*).

*Al-Azzam & Shatnawi (2021)* evaluated and compared the effectiveness and precision of both conventional semi-supervised and supervised machine learning methods for breast cancer prediction. The evaluation was conducted using the Wisconsin Diagnosis Cancer dataset for training and testing the models. Using only half of the training data, the semi-supervised algorithms produced very high accuracy (90%–98%). The most accurate models were the k-NN model for supervised learning and the logistic regression model for semi-supervised learning (*Al-Azzam & Shatnawi, 2021*).

*Boeri et al. (2020)* carried out research on the preliminary evaluation of how machine learning can be applied to predict outcomes related to breast cancer. Out of 1,021 patients who had breast cancer surgery, 610 were selected for the study. They built classification models for three categories and implemented two types of machine learning approaches, namely an artificial neural network and a support vector machine. These models achieved an accuracy range between 95.29% and 96.86% (*Boeri et al., 2020*).

*Lou et al. (2020)* conducted an analysis using machine learning methods to assess the predictive accuracy of models and determine key factors linked to recurrence within a decade following breast cancer surgery. The data from a patient registry were divided into a training set with 798 patients, an internal validation set of 171 patients, and an external validation set with 171 patients. Their results demonstrated that variables including demographic data, clinical attributes, quality of care, and preoperative life quality were closely linked to the probability of recurrence within ten years following surgery (*Lou et al., 2020*).

## MATERIALS AND METHODS

The methodology of this study is based on combining ensemble learning and deep learning algorithms using the stacking ensemble method on a breast cancer dataset to develop a predictive model for breast cancer diagnosis. The flow chart depicting the sequential tasks and processes that constitute our methodology is provided in Fig. 1. This diagram offers a detailed visual representation of each step in our methodology, highlighting the systematic progression from data collection and preprocessing to model training, evaluation, and validation. The integration of ensemble and deep learning models through the stacking method aims to increase diagnostic accuracy by leveraging the strengths of both approaches. Initially, we present the experimental dataset, predictive models, and performance metrics

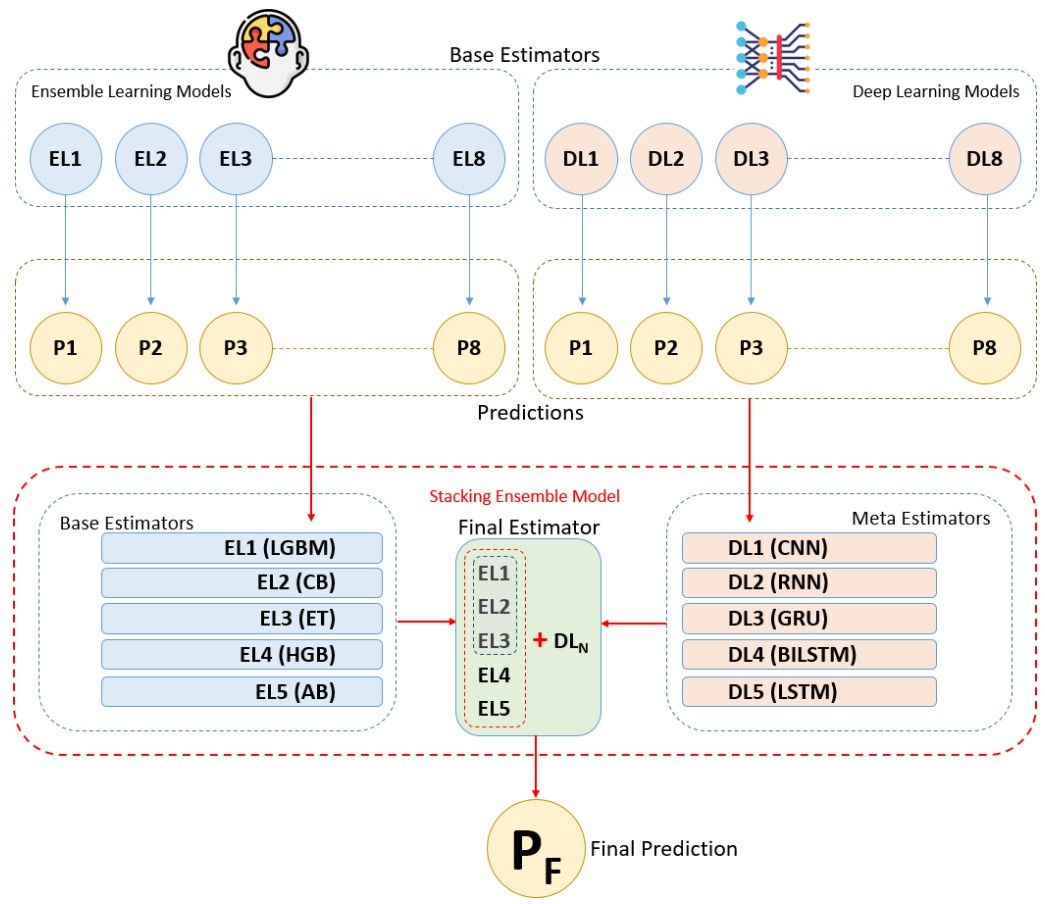

**Figure 1** Workflow and process diagrams for the proposed methodology.

used in the study. In the following section, we explain the proposed methodology and the experimental environments and tools we used step by step under the heading "Proposed Methodology and Experimental Setup".

## Data description, preparation, and feature engineering

In this study, the Breast Cancer Wisconsin (Diagnostic) dataset was chosen as the dataset for breast cancer diagnosis (*UCI Machine Learning, 2023*). This open-source dataset has been imported from a data dump file. This dataset consists of 569 samples, each with 32 features. All feature values are re-encoded with four significant digits. There are no missing feature values in the dataset. The distribution of the target classes divides into 357 (63%) benign and 212 (37%) malignant cases. Detailed information about all variables, including their types and descriptions, can be obtained from the data source provider's website (*UCI Machine Learning, 2023*). This dataset is widely used for breast cancer diagnosis and research and is considered a reliable reference source (*Ak, 2020*; *Al-Azzam & Shatnawi, 2021*; *Swassen & Okba, 2022*; *Rasool et al., 2022*).

The features in the dataset include biopsy results such as nucleus size, texture, and cell shape. Each sample is labeled as benign or malignant. To prevent features in the dataset from having different scales, which could negatively impact the model's performance, all features were standardized using standard scaler (*Aurélien, 2019*; *Nelli, 2023*; *Scikit-learn, 2024*). The standardization process adjusts each feature to have a mean of 0 and a standard deviation of 1, bringing the data closer to a normal distribution. This process facilitates faster and more accurate learning by the model (*Raschka, Patterson & Nolet, 2020*; *Scikit-learn, 2024*). Standardization is particularly essential for enhancing the performance of gradient-based algorithms and accelerating the training process (*Pedregosa et al., 2011*; *Aurélien, 2019*; *Gurcan et al., 2022a*). The dataset was split into training and test sets to evaluate the model's performance on an independent test set. The dataset is split into 10 folds using Stratified K-Fold. This splitting is necessary to assess the model's ability to generalize (*Pedregosa et al., 2011*; *Ferroni et al., 2019*; *Raschka & Mirjalili, 2019*; *Gurcan, 2023a*). The training set is employed to develop and fine-tune the model, while the test set is used to gauge the model's ability to generalize to new data and evaluate its overall effectiveness (*Al-Azzam & Shatnawi, 2021*).

In the feature engineering stage, the importance score of 30 descriptive features (independent variables) in the dataset for cancer diagnosis was extensively evaluated using ensemble models such as LightGBM (LGBM), CatBoost (CB), XGBoost (XGB), Gradient Boosting (GB), and Random Forest (RF) (*Pedregosa et al., 2011*; *Scikit-learn, 2024*). Ensemble models enhance overall accuracy and reliability by combining predictions from multiple base models. For each feature, these models measure its discriminative power during the construction of decision trees. To standardize the importance weighting of features for each model, min-max scaling was used. In this way, all importance scores were standardized between 0 and 1 (*Aurélien, 2019*; *Gurcan, 2023b*; *Nelli, 2023*; *Scikit-learn, 2024*). Then, the average of the scores obtained from all models was taken and the features were ranked according to this value (average importance score). This helps determine which features are more decisive for cancer diagnosis and which ones are less effective (*Abbas et al., 2021*; *Ebrahim, Sedky & Mesbah, 2023*).

## Identification and evaluation of ensemble learning algorithms

Ensemble algorithms are methods designed to enhance model performance by combining multiple machine learning models. These approaches aim to achieve higher accuracy, generalization, and robustness compared to individual models (weak learners). Ensemble methods capitalize on model diversity and error reduction to improve predictive outcomes. In this study, eight ensemble learning algorithms were selected and trained, with each algorithm individually trained on the training data (*Aurélien, 2019*; *Fatima et al., 2020*; *Ebrahim, Sedky & Mesbah, 2023*). The ensemble algorithms used in this study are described as follows:

- **CatBoost (CB):** CatBoost is a gradient boosting algorithm that works particularly well with categorical variables. It automatically handles categorical variables, speeding up the training process and enhancing performance.

- **LightGBM (LGBM):** LightGBM is a fast and high-performance gradient boosting algorithm, particularly effective with large datasets. It is preferred for large-scale data analysis due to its low memory usage and support for categorical variables.
- **XGBoost (XGB):** Known as extreme gradient boosting, XGBoost is an effective gradient boosting algorithm in terms of both speed and performance. It can model complex relationships and interactions using a tree-based modeling approach.
- **Random Forest (RF):** Random Forest is an advanced ensemble method that aggregates multiple decision trees to enhance prediction accuracy and reduce the risk of overfitting.
- **Extra Trees (ET):** Similar to Random Forest, Extra Trees is an ensemble algorithm that creates decision trees with additional randomization. By selecting random split points, it speeds up the training process and often yields effective results with large datasets.
- **HistGradientBoosting (HGB):** HistGradientBoosting is an algorithm that provides fast gradient boosting on large datasets. It significantly reduces training time using a quick histogram-based gradient computation method.
- **AdaBoost (AB):** AdaBoost operates as an ensemble technique by aggregating multiple weak classifiers to form a robust and effective overall classifier. It focuses on difficult examples by increasing the weight of misclassified instances, thereby improving overall performance.
- **GradientBoosting (GB):** GradientBoosting is an ensemble algorithm that attempts to model complex relationships and interactions by creating a series of weak learners. Each learner corrects the errors of the previous ones, reducing overall error.

These algorithms reveal various features and relationships in the breast cancer dataset using distinct techniques and optimizations. Each was evaluated for performance, and the top five with the best results were chosen as base learners. These base learners were integrated _via_ the stacking method to form the final predictive model. Initial predictions from the base learners are refined by a meta-learner, selected based on overall performance and generalization ability to ensure a robust and stable model (_Raschka & Mirjalili, 2019_; _Fatima et al., 2020_; _Karadeniz, Tokdemir & Maraş, 2021_; _Naseem et al., 2022_).

## Identification and evaluation of deep learning algorithms

Deep learning is an advanced subset of machine learning that uses artificial neural networks (ANNs) to automatically learn complex data structures and relationships. Deep learning models consist of multiple layers of ANNs, where each layer represents different features of the data. Deep learning algorithms are sophisticated modeling techniques particularly effective in large and complex datasets (_Aurélien, 2019_; _Fatima et al., 2020_; _Chugh, Kumar & Singh, 2021_; _Din et al., 2022_; _Rasool et al., 2022_; _Gurcan, 2023a_). In this study, the following eight different deep learning algorithms were selected and trained:

- **Convolutional Neural Network (CNN):** Effective in image processing and pattern recognition tasks. It consists of different convolutional and pooling layers to automatically extract features from input data, particularly used for detecting structural features in 2D images.

- **Recurrent Neural Network (RNN):** Performs well on sequential data such as time series. It processes sequential data by retaining information from previous steps, useful in tasks like language modeling and time series predictions.
- **Gated Recurrent Unit (GRU):** A simpler version of LSTM designed for faster training times and reduced memory consumption in RNNs, aimed at reducing overfitting in recurrent structures.
- **Bidirectional LSTM (BILSTM):** Utilizes both past and future time information to enhance time series predictions. Particularly successful in natural language processing and time series data.
- **Long Short-Term Memory (LSTM):** Effective in data with long-term dependencies like time series. Preserves long-term dependencies through memory cells and prevents gradient vanishing issues.
- **Deep Neural Network (DNN):** Deep structured neural network within traditional ANNs. Contains multiple hidden layers with full connectivity between each layer, commonly used for modeling complex structures.
- **Artificial Neural Network (ANN):** Basic ANN models are constructed by numerous artificial neurons, processing input data to achieve output results.
- **Multi-Layer Perceptron (MLP):** A basic deep learning model comprises an input layer, one or more hidden layers, and an output layer. Each neuron in a layer is fully connected to every neuron in the preceding layer, and this structure is commonly used for tasks such as classification and regression.

Throughout this process, hyper parameters of each deep learning model were optimized to achieve optimal performance. Each algorithm was individually trained on the training data. From the trained eight deep learning algorithms, the top five algorithms were selected based on performance metrics and utilized as meta-learners (final estimators) in the stacking method. The meta-learner integrates predictions from base learners to generate the final outcome at a higher level (*Aurélien, 2019*; *Fatima et al., 2020*; *Chugh, Kumar & Singh, 2021*; *Din et al., 2022*; *Rasool et al., 2022*; *Gurcan, 2023a*; *Scikit-learn, 2024*).

## Stacking integration of ensemble and deep learning models

The stacking method is one of the ensemble learning techniques aimed at combining predictions from different machine learning models to construct a stronger predictive model. In this analysis, various models called base learners (in this case, five different ensemble algorithms) are trained to make individual predictions. The main idea behind the stacking method is to use a meta learner that takes the predictions of these base learners as input and processes them at a higher level to make the final prediction. This way, the stacking method leverages the combined strength of different models, often resulting in improved performance (*Aurélien, 2019*; *Karadeniz, Tokdemir & Maraş, 2021*; *Nelli, 2023*; *Scikit-learn, 2024*).

Five ensemble learning algorithms were selected and trained on the training data. These base learners generate initial predictions, which are then used by the meta learner. The outputs of these base learners are fed into selected deep learning meta learners to create the final prediction model. In this analysis, a stacked ensemble model was built by integrating

a set of base estimators from five ensemble algorithms with a meta-estimator represented by deep learning algorithm. Different deep learning algorithms were used as the meta estimator each time to enhance generalization capability and accuracy (*Pedregosa et al., 2011*; *Aurélien, 2019*; *Karadeniz, Tokdemir & Maraş, 2021*; *Nelli, 2023*; *Scikit-learn, 2024*).

The model constructed using the stacking method was trained on the training data. Throughout the training process, efforts were made to optimize the model's parameters to achieve the best performance. Regular monitoring of the model's performance during training allowed for necessary adjustments. Post-training, the model was evaluated on test data to compute performance metrics such as accuracy, ROCAUC, F1 score, and training time. These metrics were employed to measure the model's classification performance and generalization ability. Additionally, 10-fold cross-validation was performed using Stratified K-Fold, which enhances the model's generalization ability and helps prevent overfitting. An early stopping mechanism was also employed to halt model training when the performance on the validation set began to decline. This approach mitigates overfitting and excessive adaptation to the training data, thereby improving the model's generalization capability and accuracy. The results of the analysis highlighted the strengths and weaknesses of methodology, guided future improvements, and demonstrated the importance of integrating ensemble and deep learning through stacking for better breast cancer prediction (*Lou et al., 2020*; *Naseem et al., 2022*; *Nelli, 2023*).

## Definition of performance evaluation metrics

In this research, we utilized a range of performance metrics derived from the confusion matrix to comprehensively evaluate the effectiveness of our models in diagnosing breast cancer (*Islam et al., 2022*; *Gonzales Martinez & Van Dongen, 2023*; *Scikit-learn, 2024*; *Teoh et al., 2024*). These metrics serve distinct purposes in assessing different facets of model performance. Specific formulas based on the true positive (TP), false positive (FP), true negative (TN), and false negative (FN) values derived from the confusion matrix are used to generate these measures. When taken as a whole, these measures shed light on many facets of the model's efficacy in breast cancer diagnosis. (*Pedregosa et al., 2011*; *Gulli & Pal, 2017*; *Fatima et al., 2020*; *Karadeniz, Tokdemir & Maraş, 2021*). The metrics used to evaluate the models in this study are as follows:

**Performance evaluation metrics:**

- **Precision (PPV):** The proportion of true positives among all positive predictions made by the model. It reflects the accuracy of the model when it predicts the positive class.
- **Recall (Sensitivity):** Indicates the proportion of real positives that the model properly identified. It focuses on how well the model captures the positive instances.
- **Accuracy:** Measures the overall accuracy of predictions by calculating the ratio of correctly predicted instances to the total instances.
- **ROCAUC (Receiver Operating Characteristic - Area Under Curve):** Assesses the model's capability to discriminate between classes, particularly useful in binary classification tasks such as distinguishing between benign and malignant tumors.

- **F1 Score:** A fair evaluation of the model's performance in terms of both sensitivity (recall) and positive predictive value (precision) is provided by the harmonic mean of precision and recall.
- **Training time:** The time it takes for the model to train, measured in seconds. Measures the computational effort required to train the models, essential for understanding the practical feasibility of model deployment.

**Fault evaluation metrics:**

- **False negative rate (FNR):** The proportion of actual positives that were incorrectly predicted as negative. It indicates the model's tendency to miss positive cases.
- **False omission rate (FOR):** The proportion of negative predictions that are actually positive. It highlights the model's likelihood of falsely predicting negative when the actual class is positive.
- **True negative rate (TNR):** The proportion of actual negatives that were correctly predicted as negative. It reflects the model's ability to correctly identify the negative class.
- **False positive rate (FPR):** The proportion of actual negatives that were incorrectly predicted as positive. It highlights how often the model falsely predicts the positive class.
- **False discovery rate (FDR):** The proportion of positive predictions that are actually negative. It indicates how often the model's positive predictions are incorrect.

## Proposed methodology and experimental setup

The methodology we propose in this study reveals in detail how deep learning and ensemble learning methods can be combined with stacking ensemble integration. In this section, we explain the outline of the presented methodology and how each process highlighted in the methodology is implemented to the dataset in experimental settings, as follows:

- **Loading the dataset**: In this analysis, the Breast Cancer Wisconsin (Diagnostic) dataset was chosen for the breast cancer diagnosis. This open-source dataset was imported from a CSV data file.
- **Standardizing the data**: To prevent the features in the dataset from being on different scales, which could negatively affect the model's performance, all features were standardized using the standard scaler function.
- **Feature engineering**: Using ensemble models, the significance of 30 descriptive features (independent variables) for breast cancer diagnosis was evaluated.
- **Splitting the dataset**: The dataset is split into 10 folds using Stratified K-Fold, ensuring that each fold maintains the original class distribution. This approach preserves class balance within each fold, allowing the model to be evaluated on different parts of the data while handling class imbalances consistently.
- **Determining performance metrics**: The performance evaluation metrics, including precision, recall, accuracy, ROCAUC, F1 score, and training time, were used to measure the performance of the models.
- **Determining fault metrics**: The fault evaluation metrics, including FNR, FOR, TNR, FPR, and FDR, were used to measure the fault rates of the models.

- **Building and training ensemble learning models**: Eight different ensemble learning algorithms (CB, LGBM, XGB, RF, ET, HBG, AB, GB) were fitted and trained.
- **Building and training deep learning models**: Eight different deep learning algorithms (RNN, LSTM, GRU, CNN, DNN, ANN, MLP, BILSTM) were defined and trained.
- **Defining ensemble learning models as base learners**: Among the eight trained ensemble algorithms, the top five algorithms were selected based on accuracy metrics. These five algorithms were assigned as base learners in the stacking ensemble method.
- **Defining deep learning models as meta learners**: Among the eight trained deep learning algorithms, the top five algorithms were selected based on accuracy metrics. These five algorithms were used as meta learners in the stacking ensemble method.
- **Integrating base and meta learners into the stacking ensemble model:** The prediction outputs of the base learners were taken as inputs by the selected meta learners, creating the final prediction model.
- **Training the stacking ensemble model**: The model created using the stacking method was trained on the training data.
- **Cross-validation with Stratified K-Fold:** This way splits the dataset into 10 equal parts, maintaining class distributions, and uses each part as a test set, providing a more reliable model performance evaluation.
- **Early stopping for overfitting**: This mechanism was used to halt model training and mitigate overfitting when the performance on the validation set began to decline.
- **Evaluating the models**: After training, the model was evaluated on the test and validation data, and performance metrics (accuracy, ROCAUC, F1 score, and training time) were calculated.
- **Analyzing the results**: The model's performance was analyzed to assess the benefits of combining ensemble and deep learning algorithms using the stacking method.

This study was conducted using a robust computing infrastructure. The setup included an Intel i7-12650H processor, Nvidia RTX 3060 GPUs, and 16 GB of RAM. The system operated on Windows 10 64-bit. Development was carried out in Python 3.12 (64-bit) using Jupyter Notebook version 7.1.3. PyTorch, Tensorflow, and Keras frameworks were employed for deep learning modeling. Python development environments and libraries were used in this study to create a robust model for breast cancer prediction using the stacking method. Python libraries used to carry out each process of empirical analysis are as follows:

- **pandas and NumPy:** Used for data loading, data manipulation, and mathematical operations.
- **sklearn (scikit-learn):** Used for data processing, model building, and evaluation.
- **tensorflow, keras, and scikeras:** Used for creating deep learning models.
- **xgboost, catboost, lightgbm, sklearn.ensemble:** Provides advanced ensemble learning models.
- **sklearn.ensemble.StackingClassifier:** Implements the stacking method.
- **sklearn.model_selection.train_test_split:** Splits the data set into training and testing sets.

- **sklearn.model_selection.StratifiedKFold:** Applies 10-fold cross-validation to improve model generalization.
- **tensorflow.keras.callbacks:** Provides an early stop mechanism to prevent overfitting.
- **sklearn.metrics:** Provides functions to compute various performance metrics for evaluating machine learning models.

# EXPERIMENTAL RESULTS AND DISCUSSIONS

## Exploratory data analysis and feature engineering (RQ1)

This study uses the Breast Cancer Wisconsin (Diagnostic) dataset, which comprises 569 samples and 32 features (including an identifier and a target variable). Ensemble methods comprehensively evaluate the importance of features across different models included in Table 1. Ensemble models, such as Random Forest (RF), Gradient Boosting (GB), XGBoost (XGB), LightGBM (LGBM), and CatBoost (CB), enhance overall accuracy and reliability by combining predictions from multiple base models. For each feature, these models measure its discriminative power during the construction of decision trees. Table 1 extensively demonstrates the importance various machine learning models attributed to features associated with cancer diagnosis. Particularly, features such as "worst area", "worst radius", and "worst concave points" have been identified with the highest importance scores by multiple models. These features are crucial in diagnosing cancer type and prognosis, playing a critical role in guiding patient treatment processes in clinical applications. Among the features considered in the table are factors like "mean concave points", "worst texture", "worst perimeter", and "mean texture". These features are valued to varying degrees by different models, with some being more influential than others. For instance, specific features like "mean fractal dimension" may have lower importance in certain ensemble models compared to others.

Subsequently, different numbers of features are selected based on this ranking, and cross-validation scores are calculated for each selection using cross validation score. The cross-validation scores corresponding to the number of selected features are displayed in Fig. 2. This figure illustrates how the number of selected features affects the model's cross-validation performance, demonstrating how using more or fewer features impacts model accuracy. By analyzing the graph, one can determine the optimal number and specific features required to achieve the best performance. The ideal result is obtained at the point where the cross-validation score is highest on the graph. In this analysis, it was observed that the best performance is achieved when the top 11 features are selected. Because after the first 11 features, the graph curve starts to stabilize. This finding indicates that the model achieves its best performance with these 11 features without becoming overly complex due to unnecessary features.

In conclusion, these findings serve as a valuable resource for cancer diagnosis and treatment management. Understanding which features are pivotal in cancer diagnosis is crucial for developing personalized treatment strategies. The use of ensemble methods integrates the strengths of different machine learning models to achieve more accurate,

**Table 1  Importance scores of features obtained with ensemble models.**

| Rank | Feature | RF | GB | XGB | LGBM | CB | Mean |
|---|---|---|---|---|---|---|---|
| 1 | worst radius | 0.587 | 1.000 | 1.000 | 0.255 | 0.459 | 0.660 |
| 2 | worst concave points | 0.948 | 0.286 | 0.125 | 0.702 | 0.807 | 0.574 |
| 3 | worst area | 1.000 | 0.066 | 0.046 | 0.459 | 0.944 | 0.503 |
| 4 | mean concave points | 0.764 | 0.297 | 0.190 | 0.576 | 0.564 | 0.478 |
| 5 | worst texture | 0.109 | 0.079 | 0.028 | 1.000 | 1.000 | 0.443 |
| 6 | worst perimeter | 0.572 | 0.339 | 0.351 | 0.471 | 0.363 | 0.419 |
| 7 | mean texture | 0.092 | 0.063 | 0.018 | 0.561 | 0.388 | 0.224 |
| 8 | area error | 0.197 | 0.019 | 0.009 | 0.451 | 0.290 | 0.193 |
| 9 | worst concavity | 0.254 | 0.031 | 0.025 | 0.322 | 0.297 | 0.186 |
| 10 | mean concavity | 0.470 | 0.001 | 0.013 | 0.239 | 0.162 | 0.177 |
| 11 | mean perimeter | 0.478 | 0.001 | 0.000 | 0.086 | 0.234 | 0.160 |
| 12 | worst smoothness | 0.070 | 0.016 | 0.009 | 0.302 | 0.348 | 0.149 |
| 13 | radius error | 0.085 | 0.007 | 0.022 | 0.208 | 0.311 | 0.126 |
| 14 | mean area | 0.423 | 0.001 | 0.028 | 0.035 | 0.094 | 0.116 |
| 15 | mean radius | 0.236 | 0.000 | 0.004 | 0.165 | 0.088 | 0.099 |
| 16 | mean compactness | 0.066 | 0.003 | 0.006 | 0.145 | 0.242 | 0.092 |
| 17 | worst symmetry | 0.041 | 0.000 | 0.003 | 0.314 | 0.076 | 0.087 |
| 18 | texture error | 0.008 | 0.009 | 0.000 | 0.204 | 0.127 | 0.070 |
| 19 | smoothness error | 0.015 | 0.001 | 0.009 | 0.106 | 0.198 | 0.066 |
| 20 | mean smoothness | 0.039 | 0.000 | 0.003 | 0.149 | 0.103 | 0.059 |
| 21 | compactness error | 0.022 | 0.001 | 0.007 | 0.063 | 0.192 | 0.057 |
| 22 | symmetry error | 0.007 | 0.004 | 0.002 | 0.118 | 0.150 | 0.056 |
| 23 | mean symmetry | 0.006 | 0.002 | 0.001 | 0.212 | 0.055 | 0.055 |
| 24 | worst fractal dimension | 0.014 | 0.001 | 0.006 | 0.059 | 0.168 | 0.049 |
| 25 | fractal dimension error | 0.024 | 0.003 | 0.005 | 0.078 | 0.135 | 0.049 |
| 26 | perimeter error | 0.055 | 0.004 | 0.011 | 0.082 | 0.090 | 0.048 |
| 27 | worst compactness | 0.126 | 0.002 | 0.008 | 0.051 | 0.052 | 0.048 |
| 28 | concave points error | 0.008 | 0.010 | 0.016 | 0.078 | 0.079 | 0.038 |
| 29 | mean fractal dimension | 0.000 | 0.000 | 0.005 | 0.106 | 0.000 | 0.022 |
| 30 | concavity error | 0.023 | 0.005 | 0.016 | 0.000 | 0.040 | 0.017 |

reliable, and broadly applicable results. This represents a significant step forward in improving health outcomes for patients in clinical practice.

## Prediction analysis with ensemble learning models (RQ2)

In this study, we initially evaluated various ensemble learning models for their performance in predicting breast cancer, focusing specifically on accuracy and training time (see Table 2). The results presented in Table 2 highlight the performance of various ensemble learning models for breast cancer diagnosis, with a particular focus on their accuracy, F1 score, ROC AUC, and training time. The LGBM model emerges as the top performer in terms of accuracy (0.9712), F1 score (0.9784), and ROC AUC (0.9964), while also maintaining a relatively fast training time of 0.2805 s. This makes it a highly efficient and accurate option for breast cancer diagnosis. The CB model follows closely with an accuracy of 0.9702 and

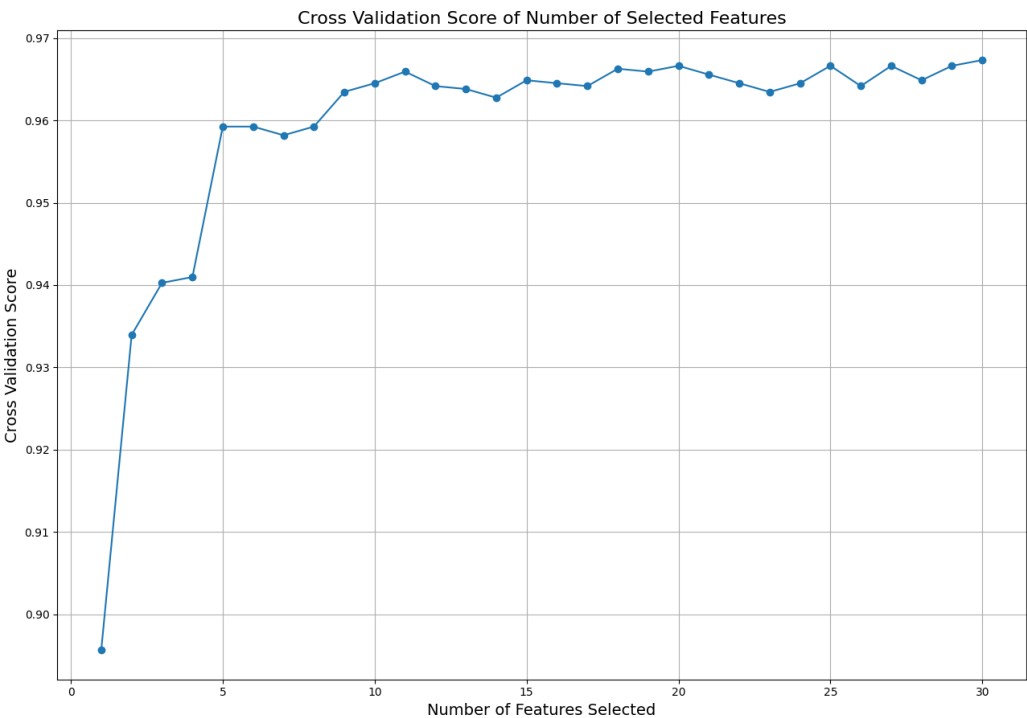

**Figure 2** Cross validation score of the selected features.

**Table 2** Performance metrics of ensemble learning models for breast cancer diagnosis.

| Model | Accuracy | F1 score | ROC AUC | Training time |
|-------|----------|----------|---------|---------------|
| LGBM | 0.9712 | 0.9784 | 0.9964 | 0.2805 |
| CB | 0.9702 | 0.9769 | 0.9900 | 0.1568 |
| ET | 0.9684 | 0.9753 | 0.9640 | 0.1125 |
| HGB | 0.9667 | 0.9742 | 0.9937 | 0.4682 |
| AB | 0.9631 | 0.9713 | 0.9934 | 0.1771 |
| XGB | 0.9614 | 0.9698 | 0.9891 | 0.0447 |
| GB | 0.9596 | 0.9686 | 0.9935 | 4.1753 |
| RF | 0.9561 | 0.9654 | 0.9894 | 1.3998 |

an F1 score of 0.9769, coupled with the fastest training time of 0.1568 s, suggesting it is both accurate and computationally efficient. ET also performs well, achieving an accuracy of 0.9684 and an F1 score of 0.9753, with one of the shortest training times at 0.1125 s. However, its ROC AUC of 0.9640 is lower than that of LGBM, CB, and others. The HGB model offers competitive accuracy (0.9667) and a high ROC AUC (0.9937), but its longer training time of 0.4682 s may limit its use in time-sensitive applications. The AB model shows strong results with an accuracy of 0.9631 and an F1 score of 0.9713, as well as a high ROC AUC (0.9934). Its training time (0.1771 s) is faster than most models, making it a solid option for efficient diagnosis.

XGB also performs well, with an accuracy of 0.9614 and a quick training time of 0.0447 s, though its ROC AUC of 0.9891 is slightly lower than the top-performing models. Both GB and RF models demonstrate lower accuracy compared to the leading models, with GB achieving 0.9596 and RF 0.9561. GB has a long training time of 4.1753 s, which may be impractical for real-time applications. Similarly, RF's training time of 1.3998 s is slower than most models, though its performance metrics remain competitive (see Table 2). In summary, LGBM and CB stand out as the most balanced models, offering both high accuracy and efficient training times. ET and XGB also present strong results, though their slightly lower ROC AUC may limit their diagnostic reliability. GB and RF, while robust, are hindered by longer training times, making them less ideal for real-time applications. Overall, LGBM and CB are the most practical and effective models for breast cancer diagnosis in this study.

## Prediction analysis with deep learning models (RQ3)

In this section, various deep learning models were evaluated for their performance in predicting breast cancer, focusing specifically on their accuracy and training time. The results presented in Table 3 indicate that the CNN model outperforms other deep learning models in terms of accuracy (0.9667), F1 score (0.9730), and ROC AUC (0.9629), making it the most effective model for breast cancer diagnosis in this study. It also exhibits the second-fastest training time (2.6082 s), suggesting it is well-suited for real-time diagnostic applications. The RNN model follows closely with an accuracy of 0.9525 and an F1 score of 0.9626, although its longer training time (3.6168 s) indicates a higher computational cost. The BILSTM and GRU models also perform competitively, with BILSTM achieving an accuracy of 0.9508 and GRU 0.9473, albeit with longer training times of 8.0068 and 6.8208 s, respectively. The LSTM model, while providing reliable results with 0.9306 accuracy, shows lower performance compared to CNN and requires a training time of 6.1488 s. The DNN, ANN, and MLP models, though slightly less accurate, demonstrate consistent performance across all metrics, with accuracies ranging from 0.9209 to 0.9245. These findings underscore the effectiveness of CNN for breast cancer diagnosis, particularly in terms of predictive power and efficiency. In comparison, while the BILSTM and GRU models offer competitive predictive capabilities, their longer training times may pose challenges for real-time applications. The LSTM model, despite its solid performance, lags behind in terms of accuracy and computational efficiency (see Table 3).

In summary, the results highlight CNN as the most suitable model for breast cancer prediction, offering an optimal balance between accuracy and training efficiency. While RNN, GRU, BILSTM, and LSTM also show strong predictive performance, their longer training times may limit their practicality in real-time applications. CNN's ability to efficiently handle complex patterns in the breast cancer dataset makes it ideal for diagnostic tasks, supporting its integration into clinical practice. Leveraging CNN in healthcare could enhance decision-making, aiding early detection and improving patient outcomes in breast cancer management.

**Table 3** Performance metrics of deep learning models for breast cancer diagnosis.

| Model | Accuracy | F1 score | ROC AUC | Training time |
|-------|----------|----------|---------|---------------|
| CNN | 0.9667 | 0.9730 | 0.9629 | 2.6082 |
| RNN | 0.9525 | 0.9626 | 0.9468 | 3.6168 |
| BILSTM | 0.9508 | 0.9609 | 0.9464 | 8.0068 |
| GRU | 0.9473 | 0.9582 | 0.9419 | 6.8208 |
| LSTM | 0.9306 | 0.9452 | 0.9227 | 6.1488 |
| DNN | 0.9245 | 0.9428 | 0.9297 | 3.1800 |
| ANN | 0.9227 | 0.9385 | 0.9267 | 3.0142 |
| MLP | 0.9209 | 0.9342 | 0.9236 | 2.8483 |

## Prediction analysis with stacking ensemble and deep learning integration (RQ4)

In this extensive analysis focused on prediction using stacking ensemble learning models, we conducted a thorough evaluation across multiple configurations of base and meta estimators, placing particular emphasis on accuracy and training efficiency. Stacking ensemble learning model combines predictions from multiple base models (base estimators) using a meta-learner (final estimator). This approach leverages the diversity of predictions generated by different base models, allowing the meta-learner to learn how to best combine these predictions to make a final prediction. Stacking ensemble is particularly effective in improving predictive accuracy and robustness by mitigating the weaknesses of individual models through their collective strengths.

The highest accuracy achieved, 0.9754, was notable when combining the ensemble models (CB, LGBM, ET, HGB, AB) as base estimators with CNN serving as the meta estimator (see Table 4). It also recorded the best F1 Score of 0.9805, indicating a well-balanced relationship between precision and recall, essential for handling any class imbalance in the dataset. Additionally, this model had an ROC AUC score of 0.9728, signifying its excellent ability to distinguish between the positive and negative classes. This configuration not only demonstrated robust predictive capability but also exhibited the most efficient training time of 20.2027 s, highlighting its computational efficiency and practical applicability in real-time diagnostic settings. Conversely, integrating the same ensemble models with more complex recurrent neural network architectures—namely BILSTM, GRU, LSTM, and RNN—yielded competitive accuracies ranging from 0.9314 to 0.9596. However, these configurations required longer training durations, spanning from 21.4708 to 32.6735 s. Despite their longer training times, these models still maintained strong predictive performance, underscoring their potential in scenarios where computational efficiency is less critical than maximizing accuracy (see Table 4). Furthermore, Table 4 provides a comprehensive overview of the fault evaluation metrics for various stacking models. Alongside the performance metrics, the CNN-based stacking model achieves the lowest FNR and FOR, signifying better accuracy in identifying true positives and minimizing errors in negative predictions. On the other hand, the LSTM-based model records the highest FNR and FOR, indicating a greater frequency of missed

**Table 4   Performance and fault metrics of stacking ensemble models (5+1) for breast cancer diagnosis.**

| Performance evaluation metrics | | | | | | |
|---|---|---|---|---|---|---|
| Stacking Model | Precision | Recall | Accuracy | F1 score | ROC AUC | Training time (s) |
| (CB+LGBM+ET+HGB+AB)+CNN | 0.9783 | 0.9833 | 0.9754 | 0.9805 | 0.9728 | 20.2027 |
| (CB+LGBM+ET+HGB+AB)+BILSTM | 0.9608 | 0.9777 | 0.9596 | 0.9682 | 0.9535 | 32.6735 |
| (CB+LGBM+ET+HGB+AB)+RNN | 0.9589 | 0.9748 | 0.9578 | 0.9666 | 0.9520 | 21.4708 |
| (CB+LGBM+ET+HGB+AB)+GRU | 0.9458 | 0.9776 | 0.9490 | 0.9607 | 0.9391 | 26.3012 |
| (CB+LGBM+ET+HGB+AB)+LSTM | 0.9241 | 0.9720 | 0.9314 | 0.9470 | 0.9178 | 26.0821 |

| Fault evaluation metrics | | | | | |
|---|---|---|---|---|---|
| Stacking Model | FNR | FOR | TNR | FPR | FDR |
| (CB+LGBM+ET+HGB+AB)+CNN | 0.0167 | 0.0276 | 0.9623 | 0.0377 | 0.0217 |
| (CB+LGBM+ET+HGB+AB)+BILSTM | 0.0223 | 0.0348 | 0.9293 | 0.0707 | 0.0392 |
| (CB+LGBM+ET+HGB+AB)+RNN | 0.0252 | 0.0427 | 0.9292 | 0.0708 | 0.0411 |
| (CB+LGBM+ET+HGB+AB)+GRU | 0.0224 | 0.0391 | 0.9007 | 0.0993 | 0.0542 |
| (CB+LGBM+ET+HGB+AB)+LSTM | 0.0280 | 0.0513 | 0.8636 | 0.1364 | 0.0759 |

positive cases and incorrect negative predictions. Additionally, the GRU-based model has a higher FPR and FDR, suggesting it tends to misclassify negatives as positives more frequently.

In addition to this analysis, we present here another stacking model that includes a different combination of ensemble and deep learning models. The stacking models presented here represent a comprehensive exploration of combining ensemble learning with diverse meta estimators to optimize breast cancer prediction. The stacking model demonstrated here combines the three best ensemble learning models with one of the five deep learning models that we chose as the top meta-predictors (see Table 5). Among these configurations, the model (CB+LGBM+ET)+CNN emerges as the standout performer with an impressive accuracy of 0.9772. This ensemble not only achieves high accuracy but also excels in F1 score (0.9819) and ROC AUC (0.9742), indicating robust performance across key metrics essential for reliable diagnostic models. Importantly, it accomplishes this with a notably efficient training time of 15.4185 s, underscoring its suitability for real-time clinical applications where prompt decision-making is critical (see Table 5).

Conversely, integrating the same base estimators (CB, LGBM, ET) with recurrent neural network architectures (GRU, RNN, BILSTM, LSTM) demonstrates varying levels of performance. While these configurations maintain competitive accuracies ranging from 0.9473 to 0.9613, they require longer training durations spanning from 18.1146 to 31.4356 s. Table 5 presents the fault evaluation metrics for different stacking models. Among the models, the CNN-based stacking model demonstrates the lowest FNR and FOR, indicating a strong capability in correctly identifying positives and minimizing negative misclassification errors. In contrast, the LSTM-based model exhibits the highest FNR and FOR, reflecting a higher incidence of missed positive cases and incorrect negative predictions. The GRU-based model has the highest FPR and FDR, suggesting it more

**Table 5 Performance and fault metrics of stacking ensemble models (3+1) for breast cancer diagnosis.**

| Performance evaluation metrics | | | | | |
|---|---|---|---|---|---|
| Stacking Model | Precision | Recall | Accuracy | F1 score | ROC AUC | Training time (s) |
| (CB+LGBM+ET)+CNN | 0.9787 | 0.9860 | 0.9772 | 0.9819 | 0.9742 | 15.4185 |
| (CB+LGBM+ET)+RNN | 0.9592 | 0.9804 | 0.9613 | 0.9695 | 0.9549 | 18.1146 |
| (CB+LGBM+ET)+BILSTM | 0.9683 | 0.9693 | 0.9596 | 0.9678 | 0.9564 | 31.4356 |
| (CB+LGBM+ET)+GRU | 0.9548 | 0.9777 | 0.9561 | 0.9656 | 0.9489 | 21.9476 |
| (CB+LGBM+ET)+LSTM | 0.9520 | 0.9665 | 0.9473 | 0.9581 | 0.9409 | 24.2173 |

| Fault evaluation metrics | | | | | |
|---|---|---|---|---|---|
| Stacking Model | FNR | FOR | TNR | FPR | FDR |
| (CB+LGBM+ET)+CNN | 0.0140 | 0.0221 | 0.9625 | 0.0375 | 0.0213 |
| (CB+LGBM+ET)+RNN | 0.0196 | 0.0337 | 0.9293 | 0.0707 | 0.0408 |
| (CB+LGBM+ET)+BILSTM | 0.0307 | 0.0467 | 0.9435 | 0.0565 | 0.0317 |
| (CB+LGBM+ET)+GRU | 0.0223 | 0.0378 | 0.9200 | 0.0800 | 0.0452 |
| (CB+LGBM+ET)+LSTM | 0.0335 | 0.0521 | 0.9152 | 0.0848 | 0.0480 |

frequently misclassifies negatives as positives compared to the other models. Table 5 shows that the CNN-based stacking model achieves the lowest FNR and FOR. This indicates its superior performance in correctly identifying positive cases and minimizing the misclassification of negative cases. In contrast, the LSTM-based model has the highest FNR and FOR, suggesting it struggles more with missing positive cases and incorrectly labeling negatives as positives.

This trade-off between accuracy and training efficiency highlights the nuanced considerations in model selection for medical diagnostics. Despite the longer training times, these models retain strong predictive capabilities, suggesting their potential in scenarios prioritizing comprehensive predictive power over rapid computation (see Table 5). The versatility of stacking ensemble methods was evident in how they effectively combined diverse base learners with different meta learners to enhance predictive performance across varying complexities of breast cancer prediction tasks. This approach not only leveraged the strengths of ensemble learning in capturing diverse data characteristics but also harnessed the deep learning capabilities of meta learners to refine predictions based on intricate patterns within the data.

In conclusion, the findings underscore the efficacy of stacking ensemble learning in enhancing breast cancer prediction models, particularly when optimized with suitable meta estimators like CNN. The results provide valuable insights into balancing computational efficiency with predictive accuracy, paving the way for further advancements in clinical decision support systems aimed at improving patient outcomes through early and accurate disease detection. This study underscores the importance of methodological innovation in machine learning, highlighting stacking as a powerful technique for integrating heterogeneous models and maximizing predictive performance in clinical applications.

## Limitations and validity

Despite the promising findings and advancements demonstrated in this study regarding the enhancement of breast cancer diagnosis using ensemble models and deep learning techniques, several critical limitations and considerations should be acknowledged:

1. **Dataset dependency and representativeness:** The study heavily relies on the Breast Cancer Wisconsin (Diagnostic) dataset, which, while widely used and well-studied, primarily represents a specific demographic and may not fully capture the diversity of breast cancer cases globally. Differences in genetic backgrounds, environmental factors, and healthcare practices across different populations could impact the generalizability of the developed models. Therefore, the application of these models to datasets from other regions or populations requires careful validation to ensure robustness and reliability across varied contexts.

2. **Generalizability and external validation:** The models developed in this research are evaluated on a controlled dataset environment. To establish their real-world applicability and generalizability, further validation using external datasets from diverse sources and healthcare settings is imperative. This external validation process would help assess how well the models perform across different demographics, clinical protocols, and healthcare infrastructures, thereby enhancing their utility and trustworthiness in broader clinical practice.

3. **Feature selection and model interpretability:** The success of machine learning models crucially hinges on the quality and relevance of the features used for training. In this study, specific feature selection methods and engineering techniques were employed, which could influence the models' performance and interpretability. It is essential to thoroughly investigate the robustness of these methods and their implications in different clinical scenarios where the availability and relevance of features may vary.

4. **Ethical considerations and bias mitigation:** The deployment of machine learning models in healthcare, particularly for critical tasks such as breast cancer diagnosis, raises ethical concerns related to patient privacy, data security, informed consent, and potential biases in model predictions. The responsible integration of these models into clinical workflows necessitates adherence to rigorous ethical guidelines and regulatory frameworks.

5. **Clinical adoption and implementation challenges:** While the developed ensemble models and deep learning techniques exhibit promising performance metrics such as accuracy and efficiency in training, their adoption into clinical decision-making processes requires comprehensive validation through rigorous clinical trials. Evaluating the models' impact on actual patient outcomes, healthcare resource utilization, and clinical workflows is crucial for assessing their effectiveness and feasibility in real-world settings. Additionally, the integration of these advanced technologies into existing healthcare systems demands considerations of scalability, interoperability, and usability to facilitate seamless adoption by healthcare providers and institutions.

6. **Model performance stability:** The performance of machine learning models, including ensemble techniques and deep learning architectures, can be sensitive to variations in data quality, preprocessing methods, and model hyper parameters. Ensuring the

stability and reliability of model predictions across different scenarios and over time requires continuous monitoring and refinement of model training strategies.

7. **Cost and resource requirements:** Implementing advanced machine learning models in clinical practice involves significant computational resources, infrastructure setup, and ongoing maintenance costs. The scalability and cost-effectiveness of deploying these models in diverse healthcare settings need careful consideration to justify their integration into routine clinical workflows and ensure sustainable adoption.

In summary, while this study contributes valuable insights and potential advancements in breast cancer diagnosis through innovative machine learning approaches, addressing these aforementioned limitations and challenges is paramount. Future research efforts should prioritize comprehensive validation across diverse populations, robust ethical considerations, and practical implementation strategies to ensure the responsible and effective use of predictive models in enhancing clinical decision support systems and improving patient care outcomes.

## CONCLUSIONS

This study has provided an evaluation on the importance of integrating ensemble methods with deep learning models for breast cancer diagnosis. Our findings are derived from comprehensive analyses conducted on the Breast Cancer Wisconsin (Diagnostic) dataset. Firstly, the results obtained using ensemble methods such as Random Forest, Gradient Boosting, XGBoost, LightGBM, and CatBoost have demonstrated the critical importance of specific diagnostic features in cancer diagnosis. Particularly, features like "worst area", "worst radius", and "worst concave points" were identified with high importance scores across multiple models, playing significant roles in the diagnostic process. Additionally, by examining how the number of selected features affects the cross-validation performance, we found that the best performance was achieved with the top 11 features, highlighting the optimization of model complexity by avoiding unnecessary feature inclusion.

Subsequently, we analyzed the effectiveness of ensemble methods including Random Forest, XGBoost, LightGBM, ExtraTrees, HistGradientBoosting, AdaBoost, GradientBoosting and CatBoost in cancer diagnosis prediction. Beyond classification success, comprehensive evaluations were also conducted on deep learning models (CNN, RNN, GRU, BILSTM, LSTM). These models present significant potential in achieving high accuracy in breast cancer diagnosis. Particularly, CNN demonstrated the highest accuracy and rapid training time, making it an optimal choice for real-time diagnostic applications. Other deep learning models were evaluated in terms of their performance and training times, providing insights into their potential uses in various clinical scenarios.

Finally, we demonstrated how the integration of stacking ensemble methods with different base and meta predictors enhances breast cancer prediction models. Specifically, the combination of effective base predictors like CB, LGBM, and ET with a powerful meta predictor like CNN highlighted their potential in healthcare decision support systems, achieving high accuracy (0.9772) with reduced training times. In conclusion, this study contributes valuable insights for improving decision-making processes in breast cancer

diagnosis and management. Our findings from integrating ensemble and deep learning methods provide a significant guide for strengthening early diagnosis strategies and enhancing patient treatment outcomes in clinical applications.

Future studies can further enhance the generalizability of models by evaluating the performance of ensemble and deep learning models across diverse populations. Additionally, research integrating new biological information such as metabolic profiles or genetic data could offer a more comprehensive approach to breast cancer diagnosis. Furthermore, advancements in modeling techniques are essential, including the exploration of hybrid or optimized versions of ensemble methods and deep learning models to achieve higher accuracy and faster learning times. Evaluating these techniques in clinical settings is crucial to demonstrate their effectiveness and usability in real-world healthcare environments. Emphasis on model interpretability and reliability in healthcare applications is also critical, requiring further investigation into how model results can be interpreted and ensured for safe use by healthcare professionals. These recommendations provide a roadmap for enhancing the effectiveness and reliability of machine learning techniques in healthcare, ultimately improving their integration and application in clinical practice.

### Funding
The author received no funding for this work.

### Competing Interests
The author declares there are no competing interests.

### Author Contributions
- Fatih Gurcan conceived and designed the experiments, performed the experiments, analyzed the data, performed the computation work, prepared figures and/or tables, authored or reviewed drafts of the article, and approved the final draft.

### Data Availability
The Diagnostic Wisconsin Breast Cancer Database is available at UCI repository and Kaggle:

– Wolberg, W., Mangasarian, O., Street, N., & Street, W. (1993). Breast Cancer Wisconsin (Diagnostic) [Dataset]. UCI Machine Learning Repository. https://doi.org/10.24432/C5DW2B.

– https://www.kaggle.com/datasets/uciml/breast-cancer-wisconsin-data.

### Supplemental Information
Supplemental information for this article can be found online at http://dx.doi.org/10.7717/peerj-cs.2461#supplemental-information.

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
