# Peer review of "Enhancing breast cancer prediction through stacking ensemble and deep learning integration"

_PeerJ Computer Science, doi:10.7717/peerj-cs.2461_

## Round 0.1 · original submission · Major Revisions

Dear authors,

Thank you for submitting your article. Based on reviews' comments, your article has not yet been recommended for publication in its current form. However, we encourage you to address the concerns and criticisms of the reviewers and to resubmit your article once you have updated it accordingly. Reviewer 2 has advised you to provide a specific reference. You are welcome to add it if you think it is relevant and useful . However, you are under no obligation to include it, and if you do not, it will not affect my decision.

Best wishes,

Reviewer 1 ·

Basic reporting

Abstract:
1. Is the abstract able to convey the significance of breast cancer as a global health concern in a manner that is simple and straightforward to comprehend?
2. Are the objectives of the study articulated in a way that is easy to understand, particularly in respect to the integration of ensemble and deep learning models?
3. Is there a summary of the most significant findings included in the abstract? More specifically, does it include the performance metrics of the model that was proposed?

Introduction
1. Specifically, to what extent does the introduction make it abundantly evident that there are limitations that are now present in the processes of breast cancer diagnosis?
2. Does the methodology that has been proposed, which combines ensemble learning with deep learning models, have a full justification that can be provided?
3. When it comes to the diagnosis of cancer, does the beginning of the article provide sufficient information concerning the limitations of typical machine learning algorithms?
Background and Related Work
1. Are there any comprehensive summaries of the research that has been conducted in the past about the application of machine learning to the diagnosis of cancer that are included in the background section?
2. A considerable number of talks have taken place addressing the recent advancements in ensemble methods and the ways in which these methods can be included into deep learning models.
3. Does the section contain specific studies that have utilised methodologies that are comparable to those that are being used in the current research? This would be advantageous for the purpose of improving the background for the current research

1. Can the deep learning models that were selected (CNN, RNN, GRU, BILSTM, and LSTM) be considered appropriate in light of the metrics that are used to evaluate their performance?

2. Are there any special criteria that are specified in the article that were utilised in the selection process for these particular models?

3. Is there any kind of comparative analysis that has been done on the performance of the deep learning algorithms that have been reviewed?

Stacking Integration of Ensemble and Deep Learning Models
1. Is there a clear explanation of the stacking process, particularly with regard to the interaction between base learners and meta-learners?
2. To what extent does the stacking process entail the identification and debate of the specific ensemble algorithms that are carried out?
3. Is the section that provides an overview of the advantages that stacking integration offers in comparison to other approaches that are more conventional?

Experimental design

Experimental Results and Discussions
1. To what extent are the results provided in a manner that is not only understandable but also concise, particularly with regard to the performance indicators of the stacking model?
2. Regarding the implications of the findings for the decision-making process that takes place in clinical settings, to what extent does the discussion correctly address these implications?
3. Would it be possible to acknowledge the potential limits of the study, and would there be a chance for a conversation to take place addressing the potential areas of research that could be carried out in the future?

Validity of the findings

N/A

Additional comments

N/A

Annotated reviews are not available for download in order to protect the identity of reviewers who chose to remain anonymous.

·

Basic reporting

The author applied stacking and deep learning (DL) integration to the breast cancer Wisconsin Breast Cancer Dataset (WBC). The proposed method as good accuracy and efficiency (low training time).

* The article written with technically correct text. Clear and unambiguous, professional English is used in general, but some further improvements can be implemented: there is a duplicated section in lines 64-70.
* The article has a professional article structure and the data and Python scripts were shared and accounted for in the article.
* I suggest the authors to add to their references the article of Gonzales-Martinez and van Dongen (2023), due to the reasons described below in Section 2 (experimental design).

Gonzales-Martinez, R., and van Dongen, D. M. (2023). Deep Learning Algorithms for Early Detection of Breast Cancer: A Comparative Study with Traditional Machine Learning. Informatics in Medicine Unlocked
Volume 41, 2023, 101317

https://www.sciencedirect.com/science/article/pii/S2352914823001636

Experimental design

The experimental design properly answers the research questions and the methods are described with sufficient detail. However, I believe that in the case of cancer detection the False Negative Rate (FNR) and the False Omission Rate (FOR) are the most relevant clinical and statistical metrics to compare cancer detection algorithms, besides accuracy. Since the differences in training time between models are just seconds, I believe efficiency is not a relevant metric to compare algorithms, and rather FNR and FOR should be included as performance metrics to compare algorithms. The reasons are described below:

1) FNR measures the proportion of actual positive cases of breast cancer that are incorrectly classified as negative cases, it quantifies the rate of missed positives (Type II errors), and hence a high FNR implies late detection of anomalies.

2) FOR measures the proportion of false negative errors or incorrect omissions in a decision-making process. As FOR captures failures to detect breast cancer, it is also relevant in the comparison of machine learning and deep learning algorithms, because missing the detection of a positive condition of breast cancer can have significant health consequences for cancer patients.

Thus, I suggest the authors to include these metrics in the core evaluation of their proposed models, as in Gonzales-Martinez and van Dongen (2023):

Gonzales-Martinez, R., and van Dongen, D. M. (2023). Deep Learning Algorithms for Early Detection of Breast Cancer: A Comparative Study with Traditional Machine Learning. Informatics in Medicine Unlocked
Volume 41, 2023, 101317

https://www.sciencedirect.com/science/article/pii/S2352914823001636

I also suggest the author to use L1-L2 regularization in the algorithms to further prevent overfitting, besides early stopping.

Validity of the findings

The validity of the findings and the conclusions linked to the findings should be evaluated on the basis of the lowest FNR and FOR, and not only on the accuracy of deep learning algorithms, since, the high level of accuracy found in the paper may be indicative of overfitting problems, despite the efforts of the author to reduce overfitting through early stopping.

Additional comments

I will strongly advise to the Editor the publication of the article, when FNR and FOR are included in the article as additional metrics to evaluate the performance of the proposed models, and if additional regularization methods are considered by the author (e.g. L1 and L2 or dropout)

---

## Round 0.2 · accepted · Accept

Dear author,

Thank you for your hard work on the revision.. One reviewer did not respond to the invitation for the revised paper and one reviewers thinks that the manuscript is improved based on his/her advise and recommends the acceptance of the paper. I also see that your paper is improved and seems ready for publication.

Best wishes,

·

Basic reporting

No additional comments. The authors improved the manuscript based on my advise.

Experimental design

No additional comments. The authors improved the manuscript based on my advise, and now the paper properly addresses the cost of false negatives in breast cancer diagnosis.

Validity of the findings

No additional comments. The authors improved the manuscript based on my advise and now the findings take into account false omission rates.

Additional comments

I am thankful to the authors for improving the quality manuscript based on my advise and my recommendations.